# Home Chimney Pinwheels (HCP) as Steh and Remote Monitoring for Smart Building IoT and WSN Applications [note 1]

**DOI:** 10.3390/s23052858

**Published:** 2023-03-06

**Authors:** Ajibike Eunice Akin-Ponnle, Paulo Capitão, Ricardo Torres, Nuno Borges Carvalho

**Affiliations:** 1Departamento de Electrónical, Telecomunicações e Informática, (DETI), Universidade de Aveiro, Campus Universitário de Santiago, 3810-193 Aveiro, Portugal; 2Instituto de Telecomunicações, Universidade de Aveiro, 3810-193 Aveiro, Portugal

**Keywords:** energy harvesting, home chimney pinwheels (HCP), internet of things (IoT), LoRa, smart building, smart city, wireless sensors

## Abstract

Smart, and ultra-low energy consuming Internet of Things (IoTs), wireless sensor networks (WSN), and autonomous devices are being deployed to smart buildings and cities, which require continuous power supply, whereas battery usage has accompanying environmental problems, coupled with additional maintenance cost. We present Home Chimney Pinwheels (HCP) as the Smart Turbine Energy Harvester (STEH) for wind; and Cloud-based remote monitoring of its output data. The HCP commonly serves as an external cap to home chimney exhaust outlets; they have very low inertia to wind; and are available on the rooftops of some buildings. Here, an electromagnetic converter adapted from a brushless DC motor was mechanically fastened to the circular base of an 18-blade HCP. In simulated wind, and rooftop experiments, an output voltage of 0.3 V to 16 V was realised for a wind speed between 0.6 to 16 km/h. This is sufficient to operate low-power IoT devices deployed around a smart city. The harvester was connected to a power management unit and its output data was remotely monitored via the IoT analytic Cloud platform “ThingSpeak” by means of LoRa transceivers, serving as sensors; while also obtaining supply from the harvester. The HCP can be a battery-less “stand-alone” low-cost STEH, with no grid connection, and can be installed as attachments to IoT or wireless sensors nodes in smart buildings and cities.

## 1. Introduction

In recent times, there have been several upgrades in electrical system designs for industrial buildings and homes, especially in terms of compactness, to come up with micro/nano electromechanical devices (MEMS or NEMS), wireless sensors, industrial sensors, internet of things (IoT), and devices for the Industrial Internet of Things (IIoT). This is in order to produce more devices that are of low power consumption, are more cost-efficient, and rely on self-powering technologies [1,2]. The aim of the foregoing being to enhance the functionalities of smart buildings, which are integral components of smart cities. A smart city is any municipality that is concerned with modernisation by the use of modern communication and information technologies (IT) to acquire, process, and disseminate information; manage resources and services efficiently; and improve the standard of living of the people [3,4].In a smart city, modern technology with information technology for high speed and high data communications via the Cloud is the order of the day, with processes being carried out faster [5]. Here, things, objects, and humans communicate among themselves with ease via the Internet, and there is a need to convert physical phenomena to sensory data and transmit the same among the networks that make implementation of activities within the smart city possible.

Consequently, the unending invention of the devices for smart city implementation, especially those for IoT and IIoT, is raising concerns for how they may be continuously powered without interruption to automation and monitoring processes of the smart city that they are meant to serve. This is especially important for those IoT devices meant for continuous environmental monitoring and control; those for uninterruptible health monitoring; and those for important industrial process monitoring, bearing in mind that, by reason of location and low power requirements of these devices, connection to the grid is economically impractical or inefficient [6]. Studies have also revealed that the use of batteries cannot fully conform to their energy challenges, because this will cause the need for constant battery replacement and routine visits to load points, which will be almost impossible for inaccessible nodes and endangered zones; such as during wars or earthquakes. Moreover, there would be an accompanying increase in overall maintenance costs of the smart city by periodic acquisition of batteries [7], and there could be a problem of leakage of the battery’s internal contents, as well as the problem of disposing used batteries or recycling, coupled with the fact that disposed batteries often contain chemicals that are toxic to the environment. These effects are not desirable and can constitute an increase to the Greenhouse Gas (GHG) Emission index and its associated extreme events and negative impacts, as illustrated in Figure 1a. Hence, the motivations for STEH are shown in Figure 1b [8,9].

Meanwhile, the smart city is credited with the deployment of information technology for the day-to-day running of activities in modern cities, and several means have been explored to provide energy needed to sustain smart city implementation, some of which are considered harmful, and many practices that promote modernisation are oftentimes disruptive to the ecological system. This has encouraged many governments around the world to issue different legislation that drives consumption of green energy to enhance ecological sustainability. As a fall-out, the European Union (EU) Parliament in 2002 issued directives on energy consumption of IoT in smart buildings; which was also due to prediction by a scientist that a smart building will have over 500 connected IoTs by 2022 [10,11]. Hence, there is a need to pay close attention to smart buildings’ energy needs as well as smart city needs in general.

Therefore, for these reasons, researchers have continued to study more on how to sustain the technologies of miniaturization of devices in order to create smarter ones, and at the same time, scout for more and improved sources of energies that are renewable and self-sustaining to operate them [12]. More so, these self-powering devices are expected to operate on no-battery and/or “no energy storage” facilities, so that they can be deployed for and withstand many applications that are inaccessible or in remote locations; and they are to be self-sustaining and long lasting, with ubiquitous functionalities [6,13].

In this paper, the concerns for smart buildings/smart city energies needed for automation are addressed by looking into how to convert the immediate energy around the surrounding of buildings to electrical energy form for instant consumption by devices for smart city implementation. The broader objectives are to make the modern building environment smart and compact, and to allow more free spaces for other important activities taking place in the industry or within the building facilities. This is one of the impacts left behind by the COVID-19 pandemic and its accompanying lockdown approach, which was controlled by social distancing. It was reported in the literature by researchers in the Ref. [14] that workplace social distancing during an epidemic or pandemic, as the case may be, can result in a 23% reduction in the transmission of disease. In this way, the applications of smart devices were easily deployed during the lockdown, while a social distancing control measure was employed to combat the effects of the COVID-19 pandemic on the global world, especially in the areas of health care and industrial applications [15,16]. In addition, public health organisations (PHO) recommended the use of disruptive technologies such as IoT, IoMT, big data technologies, and artificial intelligence (AIs), to mention but a few, to control the COVID-19 pandemic in order to lessen the burden on healthcare teams, and to maintain social distancing [17]. This allowed for controlled outspread of the disease and reduced its transmission from person to person. As a result, scientists have continued to create more of these autonomous devices, and a trillion dollars of IoT and IIoT devices have been proposed to be deployed to smart cities by the end of the year 2050 [18].

Energy harvesting is the process of converting energies that are wasting away as ambient in the environment into usable sources in order to power autonomous devices, such as those for Internet of Things (IoT), and/or Wireless Sensor Networks (WSNs) [12,19,20,21,22]. Energy harvesting (EH), which is the process of converting ambient energy from the environment in the guise of mechanical, thermal, radiant or fluid flow to electrical form, has been widely reported in the literature as green sources of energy consumption to smart city autonomous devices [6,12,19,20,21,22,23,24,25,26,27,28,29,30]. Here, the mechanical sources of energy can be in the form of vibration or pressure; the radiant energy is from a solar or radio-frequency (RF) wave; while fluid flow energy is from wind and hydro. These energies that can be converted using the EH scheme could in return be used to supply power to low energy-consuming devices, such as IoT, wireless sensor networks (WSN), Bluetooth low-energy (BLE) devices, radio frequency identification (RFID) tags, and so forth. Energy harvested by these means is renewable, long-lasting, self-sustaining and ubiquitous. It is noteworthy that IoT devices are of low power consumption in the range of µwatts to milli-watts, [29], which informs the merit that energy harvesting (EH) mechanisms are sufficient to power these devices [19].

Meanwhile, there are different phenomena that relate to the study of the energy harvesting mechanism. By this, studies have shown that EH mechanisms can be naturally occurring phenomena, such as turbulence from the wind or ocean in fluid flow energy; or sunlight, as in solar energy harvesting. Additionally, EH methods can be man-made (that is, artificial), such as energy from electromagnetic radiation or artificial vibration sources, such as man-made motion and pressure [23,24,25,26,31,32,33,34,35].

In the Ref. [19], EH mechanisms in smart cities were presented as making the autonomous devices in modern cities scavenge for their own individual energies in close proximity to the load point, since different ambient energies are usually available in the immediate environment in the form of mechanical vibration, radiant, heat or fluid energy. This is with the objective that energy supply to smart city autonomous devices will become more accessible with reduced cost as against acquiring energy from grid-sources which are associated with accompanying huge investments on grid set-up, transportation, or purchase of cables and related accessories. Besides, green energy will be available to inaccessible WSN or IoT nodes functioning in war zones or in the situation of a pandemic breakdown, such as in COVID-19 lockdown scenarios. This also agrees with the UN sustainable developmental goals of 2050, for the protection of the environment and ecology preservation. Energy harvesting mechanisms in smart cities and its impact on the smart city in modern times have been well-presented in the literature, with the IoT, WSN and IIoT being major areas of application. Here, they are an interconnected network of things, objects, or humans exchanging data among themselves via the Cloud, with the aid of available automation devices. Hence, the devices are said to be smart when objects-to-objects; humans-to-humans; humans-to-things-to-objects; and things-to-things are communicating among themselves [6,27].

This paper was therefore borne out of making IoT and IIoT devices to harness self-sustainable energy from the environment, and with the additional function/advantage of making the IoT devices an autonomous monitoring source to the energy harvesting system from which they are deriving energy. This is made possible by the use of Smart Turbine Energy Harvesters (STEH) to harvest renewable energy from ambient sources, which is wind in this case, in order to supply IoT and autonomous devices of smart buildings, while at the same time monitoring the energy data of the environment by the use of Cloud technology. In Cloud technology, data and information of different networks are allowed to be stored and retrieved at minimal or no cost depending on the bandwidth consumption of the end users. Cloud technology and computing present the merit of large-scale computations in a centrally managed Cloud-based environment [36,37]. In this study, the “thingspeak for IoT”, which was developed as an analytic platform service from mathworks, is used as a Cloud server and a "proof-of-concept IoT network". Thingspeak allows data to be visualised at an instant, with an online processing and analysis of the STEH data as soon as it arrives on the platform.

Therefore, in this study, the interconnected network of energy data in the immediate environment is in one aspect serving as an example of an automation system of the smart city via the Cloud, while the energy harvesting procedure of the entire system is, on the other hand, serving as a smart city mechanism of an energy harvesting solution for the purpose of powering up the system. In this scenario, the use of MCU ESP32 was employed and integrated with a LoRa transmitter and receiver to transmit and communicate the energy data, thereby making "energy-speak" the IoT aspect of the entire set-up. This is thereby fulfilling the objectives of the Energy Harvesting Mechanism in a smart city by making autonomous devices scavenge their own individual energy in close proximity of consumption [19].

A STEH can either be of a smart wind turbine energy harvester (SWDT-EH), or smart water turbine energy harvester (SWRT-EH), hence SWDT-EH (which is a smart wind turbine energy harvester) and STEH are used interchangeably in this paper.

In this work, we propose a smart wind turbine harvester from home chimney pinwheels (HCP-STEH), also known as a Roof Turbine Ventilator, to supply power to low-power IoT devices in buildings, WSNs, and other autonomous components in a smart city. The device is lightweight with low inertia, sensitive to wind, and is common in cities and rural places. It can serve as a cost-effective STEH to operate low-power smart city autonomous devices. It is further proposed to make the IoT devices connected to the harvester to serve as devices by which the energy harvester output is observed and monitored via the Cloud by means of transmitting and receiving devices while also obtaining supplies from the harvester.

Various researchers have investigated the use of roof turbine ventilators to generate electricity with different methods of energy conversion, as well as for divers’ purposes, as reported in the literature. The use of electro-active materials, electromagnetic converters, and piezo-converters, with different methods of integration and construction, have been demonstrated in some previous studies.

Researchers in the Ref. [38] used multiple pieces of electro-active material actuators as the energy converter via a gear mechanism with a roof turbine ventilator. The electro-active materials are attached around the base of the ventilator which impacts and vibrates by the rotation of the gear teeth attached to the bottom frame of the ventilator. Here, the use of up to six electro-active materials connected in a parallel circuit design, which yielded an amount of current up to 16.41 μA and power of 45.23 μW, was an improvement on the use of a single electro-active material with 5.41 μA and 13.68 μW of current and power, respectively. Dangeam in 2011 [39] added a three-phase synchronous generator to a roof ventilator. The construction contains a low flux density permanent magnet in the rotational part and a three-phase winding stator installed at the ventilator base. The generated AC voltage was rectified to DC and passed to a battery charger charging a 12 V 5 A lead-acid battery. In like manner, Daut et al. [40] carried out an investigative study on a modified roof ventilator to generate electricity by adding extra fins to help it spin faster than usual. The system combines a roof ventilator, AC generator, solar charger, batteries and an inverter. This system was proposed to charge a 12 Vdc backup. Other researchers, such as Hsieh et al. in 2013; Torasaa and Sermsri in 2015; and Lau et al. in 2018 [41,42,43], studied the use of rooftop ventilators for the conversion of wind to electrical energy either for domestic or commercial purposes. Results from some of the studies indicated that the set-up could not produce enough electric energy for domestic usage.

However, in this work, the use of a brush-less dynamo as the energy converter through mechanical coupling by contact with the HCP is presented, with the output being rectified and passed to a power management unit, which then supplies power to the connected IoT device for the purpose of remote monitoring of the output via the Cloud.

The remaining parts of this paper are arranged as follows: Section 2 presents the structural description and dynamics of the HCP SWDT-EH, and its suitability for harvesting wind energy from rooftops of buildings. It briefly discusses aerodynamic forces of wind energy that is incident on the HCP. It also presents the relevance of rooftop shapes’ architecture to the HCP when used as VAWT for STEH. In Section 3, the model, fabrication and experimental analysis of the HCP SWDT-EH in the laboratory are presented. Additionally, the experimental setup of the HCP-STEH, both in the laboratory, and at the rooftop, and its observation and monitoring via the Cloud (Thingspeak analytic platform) using transmit/receive LoRa devices are presented. Results obtained and corresponding discussions are presented in Section 4. Section 5 concludes the paper with some other recommendations.

The contributions of this research are thus highlighted:-Contributions to smart buildings energy requirements.-Enhancement of smart city automation networks.-Self-sustainability of devices for the smart city.Proof-of-concept IoT for smart buildings.-Promotion of social distancing at work places and industrial buildings.-Reduction in climate risk extreme events, and other GHGE impacts.

## 2. Structural Description and Dynamics of SWDT-EH

The structural dynamics of HCP relating to its suitability to function as a smart turbine energy harvester (STEH) is presented in this section. Basically, the structural dynamics of the wind turbine can be considered under the action of three main contributing structures, namely: the rotating blades, the rotor shaft, and the tower height. In the case of the HCP, the tower height has to do with the rooftop, since the HCP does not possess any physical tower, but it is commonly mounted on the chimney stack. Hence, in the use of the HCP, the advantage of the rooftop altitude is taken to harness wind energy. Therefore, the rooftop shapes are discussed briefly in the place of tower height.

### 2.1. The HCP as Vertical Axis Wind Turbine (VAWT) on Rooftops

Turbines are usually presented in vertical or horizontal orientations depending on the axial direction of their rotor shaft. Many scientists have reported about turbine harvesters in various categories to convert wind or water energy to electrical forms. In addition, the form of turbine harvesters used on rooftops is usually the VAWT, which researchers in the Ref. [20] reported to be more advantageous than the horizontal turbine in terms of simplicity of installation and energy delivery. The conventional VAWT and its multiple-blades form is as illustrated in Figure 2.

The HCP in this study is a device with a round vent, and in the shape of a circular fan which is sensitive to wind. Generally, the pinwheels of the HCP is a multiple ellipse-shaped blade turbine, which can be categorised as a multiple-blade VAWT. Each blade is terminated at a circular hub with a fixed knob at the top, and they freely span to join a circular base at the bottom. When wind blows on it, the fan rotates and in turn ventilates air up through the vent. The chimney, though being one of the most desirable features in any new home, must however be well-designed for efficient delivery and for safety purposes [44,45]. The HCP is passively positioned to serve as an external cap to the home chimney stack, as shown in Figure 3.

### 2.2. Aerodynamics of Wind on HCP for SWDT-EH

Wind energy harvesting can be studied based on the dynamic phenomena of wind turbulence, such as wake galloping, divergent oscillation, vortex-induced vibration, or rain vibration. In addition, certain static activities, such as over-turning, static instability, static deformation, and sliding are some of the other common phenomena of wind vibrations for energy conversion. These phenomena have been reported in the Ref. [23], where the researchers investigated the aerodynamic phenomenon of wind turbulence using wake galloping. In their study, they experimentally verified and proposed that this phenomenon can be a good approach to wind energy harvesting for low power equipment in the situation when the conventional turbines could be too costly and inefficient. This application was proposed for light poles and structural health monitoring devices.

The dynamics of the HCP can be analysed in the context of fluid flow at low and moderate wind speeds. The HCP can be represented as a bluff-body, whereby certain aerodynamic forces from the wind act on the pinwheels (blades or vanes) to produce mechanical vibration, which results in its continuous spinning. When the body is mounted on the roof to cover a chimney, and a beam of wind is incident on the blade, the aerodynamic forces of lift and drag act on the blade. The aerodynamic force acting on a single blade as a lift and drag is illustrated in Figure 4a,b [46]; it is the simplified view of the mid-horizontal cross-section of the turbine.

Blade angle (or blade surface orientation) is the angle between the blade and the radial direction from the vertical axis of the turbine. Since the blades are terminated at a hub at the top, and a circular base at the bottom, therefore rotation of the rotor shaft of the HCP is considered to result from the aerodynamic force of lift from the wind energy incident on the HCP blades. The force of lift from the wind, which is affected by the blade angle, must be higher than the drag for rotation of the pinwheels to occur. As the pinwheels rotate, the blade’s surface or tip moves slower to the wind. The speed differential provides force that drives the blade. The blade angle must neither be too small nor too large.

The power delivered by the STEH to overcome drag under the action of a single blade, is a function of the aerodynamics of the turbine, and this is related to the average speed delivered by wind energy to the blades of the HCP, as represented by Equation (Equation 1):(1)$$P_{STEH}=\frac{1}{2}\rho C_{p}AV_{d}^{3}$$
where, $P_{STEH}$ is the wind power, ρ is the air density (Kg/m^3^), A is the swept area of the blades (m^2^), $C_{p}$ is the drag coefficient, and $V_{d}$ is the speed differential between the actual wind speed and the speed on the blade’s surface (blade tip) in (m/s).

The drag coefficient depends on the shape of the blade, as well as whether it is straight or curved. Hence, the power of the STEH increases with the cube of wind speed. Attaching many pinwheel blades will produce more power, increased rotational speed, and even out vibrations, while the circular base is also considered to be acting as a flywheel.

Aerodynamic forces are usually calculated using some of the blade element method (BEM), and its extension to the unsteady environment [38]. The one-dimensional approximation of structures is mostly adopted as models for structural modeling of a flexible component of the wind turbine. Additionally, first-order linearised modal-based methods were used in many state-of-the-art models to represent small and moderate deflections.

### 2.3. Rooftop Shapes and Propensity of Reception for Wind Energy

It is a common believe in modern times that rooftops make up to 25% of smart city surfaces, hence, attention is presently being shifted to roof shapes in smart cities for efficient energy delivery, optimization, and consumption. To this end, there have been many studies reported about different topologies of roof shapes and design all over the world, and their importance and heights relative to the potential of wind energy harvesting on rooftops [47,48,49,50,51].

In general, there are three common types of roof shapes, namely: Gable, Hip, and Flat. A combination of these roof shapes is also possible. In most Mediterranean cities, rooftops are made of red brick terracotta-clay tiles with a distinct red colour. The durability and unique style of these building roofs make them an excellent choice to harvest energy from wind using the SWDT-EH. However, not all roof shapes mentioned above are suitable for SWDT-EH for the reasons explained in the following itemized list:Gable: A gable roof has two sections whose upper horizontal edges meet to form its ridge. It is the most common roof shape in cold or temperate climates, and it is ideal for terrains with heavy rainfall and snow. However, though its high height could have been an added advantage in the application of SWDT-EH, this roof style is not usually recommended for areas with high winds, because of its usual over-hanging shape from the upper face of the house, which may cause the roof to peel off, and easily blown away in the situation of very high wind or hurricanes.Hip: A hip roof usually slopes on all sides without any vertical side. It has strong features for high wind and snowfall because of its inward slope on all sides. It is a good design for improved ventilation. This type of roof is popular in Mediterranean regions, and they may be combined with gables in most cases. This type of roof shape may be used with SWDT-EH.Flat: This type of roof is low-pitched and flat. The choice of flat roof was adopted in this work because of its advantage to be able to withstand wind and other stormy actions more than others. In addition, it provides easy access to the rooftop than other types.

In the Ref. [50], a systematic measurement of wind characteristics was carried out on flat rooftops of tall buildings, using wind tunnel testing, while examining such building parameters as height and width ratio, relative to the disposition of such building for wind energy harvesting potential. These researchers submitted that at various locations and different wind directions, the vertical wind profiles’ patterns on rooftops were similar. They however added that, for turbines mounted on flat rooftops of rectangular buildings, a 45° wind direction was the most favourable to harvesting wind energy.

## 3. Fabrication, Set-Up, and Performance

The stages of the energy harvesting mechanism for the wind HCP-STEH through energy conversion processes up to the load point which is the IoT nodes/WSNs is represented by the block diagram shown in Figure 5. The first block in broken lines represents the HCP-STEH, the second block is the power management section, the third block is the load, and the fourth block is the Cloud for remote monitoring of the HCP-STEH. The STEH output is remotely monitored via the Cloud by means of transmitting and receiving devices connected through LoRa while also obtaining supply from the same STEH.

### 3.1. The SWDT-EH from HCP

The energy conversion process of the wind STEH in the first block of Figure 5 is divided into two stages:(i)The conversion from wind energy to mechanical energy by rotation; and(ii)Conversion from mechanical energy to electrical energy through electromagnetic induction.

To achieve (i), the home chimney pinwheels adapted as the STEH is an 18-blade ellipse-shaped roof ventilator turbine, each blade being 30 cm in length, and is terminated at a 12 cm diameter circular hub with a fixed knob at the top. These blades are meant to spin effortlessly in the wind, and vibrate the adjoining circular base of 70 cm circumference at the bottom, which also freely rotates in return with the blades.

To actualize (ii), an electromagnetic converter is required. Since the HCP-STEH is of very low inertia, and is light-weight, a small light-weight "off-the-shelf DC out-runner brush-less motor" was adapted to perform the function of an electromagnetic converter, which in this case is acting as a dynamo. The brush-less dynamo has a high power-to-weight ratio, high efficiency, low maintenance and is of low cost. The outer rotor assembly is made up of a cylindrical permanent magnet attached to a yoke with a spindle at the center, and the outer part of the yoke is locked to a cylindrical plastic covering, which altogether make up the magnetic cap of the converter having a top diameter of 3 cm, and a side length (height) of 1.6 cm. The fixed stator is made of three coil windings with three terminals, and here, two of the terminals are used for the output readings. The dynamo was mechanically fastened to the HCP through a harness which makes the rotor assembly (that is, the magnetic cap) able to be in contact with the circular base of the HCP. This ensures that the rotary motion of the HCP is transmitted to the rotor. The harness was coupled with the aid of a fabricated assembly made by a 3-D printer. The gain in speed of the rotor of the electromagnetic converter is related to the ratio of the diameter of the circular base of the HCP to the diameter of the rotor assembly. The complete fabricated HCP-STEH is as shown in Figure 6 [46].

The rotation of the pinwheels with the rotor from the top view is as illustrated in Figure 7. This indicates that at the mid-horizontal cross-section of the turbine assembly, the edges of the blades’ surface protrude out beyond the circular base. Here, $R_{p}$ is the radius of the circular base of the pinwheels; $R_{b}$ is the radius of the circle traced out by the edge of the blades surface at the mid-horizontal cross-section of the turbine; $\omega_{p}$ is the angular speed of the rotating wheels; $V_{p}$ is the linear speed at the edge of the circular base; $V_{b}$ is the linear speed at the edge of the blades surface; and $V_{wind}$ is the wind speed. The blade edge speed stems from the lift force acting on the blades in the direction transversal to the blades, and since the blades are free to move transversely, this thereby causes the rotation of the blades about its central vertical hub. The blade edge speed will usually be less than the wind speed. $R_{d}$ and $\omega_{d}$ are the radius and the angular speed of the rotor assembly, respectively.

When a beam of wind is initialized, an inflow of wind energy strikes the turbine blades and makes the turbine rotate, or spin. This in turn transmits rotation to the attached rotor of the electromagnetic converter, thereby generating voltage across its output by inducing emf in the stator windings.

The linear speed at the edge of the blade surface $V_{b}$ is related to $R_{b}$, as,
(2)$$V_{b}=\omega_{p}R_{b}$$
Likewise, the linear speed at the edge of the circular base is related to the radius of the HCP base as,
(3)$$V=\omega_{p}R_{p}$$
substituting for $\omega_{p}$ from (2), then (3) becomes,
(4)$$V=\left(\frac{R_{p}}{R_{b}}\right)V_{b}=k_{b}V_{b}$$
Usually, $R_{p}$ is taken to be lesser than $R_{b}$, hence, $k_{b}$ < 1. In addition, neglecting friction losses, the linear speed at the edge of the circular base $V_{p}$, will also be the same at the edge of the rotor assembly of the electromagnetic converter.
(5)$$V=\omega_{d}R_{d}$$

Equations (3) and (5), and rearranging,
(6)$$\omega_{d}=\omega_{p}\left(\frac{R_{p}}{R_{d}}\right)$$
Let $k_{r}=\left(\frac{R_{p}}{R_{d}}\right)$, be the ratio of the radius of the circular base to the radius of the rotor (that is, gain in rotor speed), therefore,
(7)$$\omega_{d}=k_{r}\omega_{p}$$
Substituting for $\omega_{p}$ from (3), then (7) becomes
(8)$$\omega_{d}=k_{r}\left(\frac{V}{R_{p}}\right)=2\pi N_{sd}$$
where, $N_{sd}$ is the angular frequency of the rotor (in rps).

From (8),
(9)$$N_{sd}=k_{r}\left(\frac{V}{2\pi R_{p}}\right)$$
For the electromagnetic converter, adopting the general emf equation of a dc generator,
(10)$$E_{g}=k_{c}\phi N_{sd}$$
where $E_{g}$ is the generated emf, ϕ is the magnetic flux, and $k_{c}$ is the constant of proportionality which accounts for other factors, such as number of poles, number of conductors, and so forth.

Substituting for $N_{sd}$ from (9), (10) becomes,
(11)$$E_{g}=\left(\frac{k_{c}k_{r}\phi }{2\pi }\right)\left(\frac{V}{R_{p}}\right)$$
In like manner, substituting for $V_{p}$ from (4), (11) now becomes,
(12)$$E_{g}=\left(\frac{k_{c}k_{r}\phi }{2\pi }\right)\left(\frac{k_{b}V_{b}}{R_{p}}\right)$$
For the purpose of simplification, assuming the wind speed bears a direct relationship with the linear speed at the edge of the blades as:(13)$$V_{b}=k_{w}V_{wind}$$
where $k_{w}<1$, and accounts for factors such as air density, pressure, blade swept area, and so forth, with some other factors peculiar to the blade design. Additionally, substituting (13) for Vb, (12) then becomes,
(14)$$E_{g}=\left(\frac{k_{b}k_{\omega }k_{c}\phi }{2\pi }\right)\left(\frac{k_{r}V_{wind}}{R_{p}}\right)$$
Equation (Equation 14) shows that the no-load output voltage from the developed HCP-STEH is expected to show a linear relationship with the wind speed, and the gain in rotor speed. This implies that, for this coupling arrangement of the HCP with the electromagnetic converter, the rotor diameter must be made as small as possible relative to the diameter of the pinwheels circular base in order to have high gain in rotor speed leading to high output voltage from the electromagnetic converter.

For the two coil windings used in the stator, if the generated emf in them are designated as $E_{A}$ and $E_{B}$, and assuming their waveforms are equal in magnitude and sinusoidal but, with $E_{B}$ leading or lagging $E_{A}$ with angle β.

If $E_{A}$ varies with time as:(15)$$E_{A}\left(t\right)=E_{g}e^{j\omega t}$$

Likewise, $E_{B}$ will vary as:(16)$$E_{B}\left(t\right)=E_{g}e^{j\omega \pm \beta }$$
where, β is the phase difference between $E_{A}$ and $E_{B}$.

The open circuit voltage at the two terminals is given as:(17)$$E_{o}\left(t\right)=E_{A}\left(t\right)+E_{B}\left(t\right)=E_{g}e^{j\omega t}+E_{g}e^{j(\omega t\pm \beta )}$$
(18)$$E_{o}\left(t\right)=E_{o}\left(t\right)=E_{g}e^{j\omega t}\left(1+e^{j(\pm \beta )}\right)$$
(19)$$E_{o}\left(t\right)=\sqrt{2}E_{g}e^{j(\omega t+\theta )}$$
where,
(20)$$\theta =\tan^{-1}\left(\frac{\sin(\pm \beta )}{1+\cos(\pm \beta )}\right)$$

If $E_{B}$ lags $E_{A}$ by $120^{\circ }$, then the resulting output voltage waveform, $E_{o}$, is sinusoidal and will lag $E_{A}$ by $60^{\circ }$. In reality, the waveform lies between being sinusoidal and trapezoidal. The magnitude varies directly with wind speed and gain in rotor speed as shown in (14).

### 3.2. The Power Management Unit (PMU)

The harvested electrical energy from the electromagnetic converter is rectified by diodes, and supplied to the Power Management Unit (PMU) of the harvester. The PMU is a DC–DC converter on an integrated circuit. The PMU employs the AEM30940 chip which is an integrated energy management IC that extracts DC power from a micro-generator (This could be wind or piezo), or any high-frequency RF input to simultaneously store energy in a rechargeable element and also supply two independent regulated voltages as outputs, [52,53]

The chip can operate with input voltage as low as 50 mV, and requires input power of just 3 μW for a cold start. It can convert varying input DC voltages into usable fixed DC voltage in the low-voltage and high-voltage Low Drop-Out (LDO) regulators. Configuration pins determine various operating modes of the chip, and conditions of the energy storage element, as well as the output voltage of the low-voltage and high-voltage LDO regulators. The chip contains three pins for configuring the output voltages, and two pins for configuring the Maximum Power Point (MPP) tracking ratio, which must be selected according to the type of harvester. The chip has a pin dedicated to charging the energy storage element, and another pin for a primary battery, which is optional, and was not used in this work.

Figure 8 shows the circuit diagram of the PMU, where the AEM30940 was configured for an MPP ratio of 50% and the voltage of the high-voltage LDO and the low-voltage LDO regulators are respectively 3.3 V and 1.8 V. The high-voltage LDO supplies power to the connected LoRa module, and the microcontroller (MCU) in the module is used to sense the output of the HCP-STEH. Diodes D1-D4 in bridge configuration rectifies the output of the electromagnetic converter of the HCP, and the voltage at the two ends of resistor $R_{1}$, $V_{A}$ and $V_{B}$ are read by the MCU in the LoRa module. $V_{A}$ is taken as the output voltage, and the current is estimated by considering the difference between $V_{A}$ and $V_{B}$ with the value of $R_{1}$. A dual-cell super-capacitor was used as the energy storage element.

### 3.3. Smart Sensing and Cloud Platform

The harvester PMU was integrated with an ESP32 LoRa 1-CH Gateway module which combines an ESP32-WROOM32—a programmable microcontroller unit (MCU) featuring both WiFi and Bluetooth radios—with an RFM96W LoRa transceiver [from Sparkfun, Niwot, CO, USA]. This module is supplied with power from the HCP-STEH by making use of the stored energy of the super-capacitor in the PMU. In this work, the MCU was used to read the output of the HCP-STEH and collect its voltage and current output values by means of its internal 12-bit ADCs. The LoRa transceiver in the module is then used as a transmitting device, sending the acquired data through its antenna to an identical module within the building, across a reasonable distance via LoRa technology. After the information is obtained by the receiving module, the receiver MCU relays it to an IoT Cloud platform, "ThingSpeak" from Mathworks, using a Wi-Fi connection.

In order to save power, the transmitting module was programmed to be in a low-power state (sleep mode) for the time that data update is not needed. During this state, the module takes very little current, and the current from the HCP-STEH charges the super-capacitor. For this work, a data update time interval of 15 min was set, at which the MCU is awakened from its low-power state to read the instant generated output voltage and current values of the STEH. The LoRa transceiver then sends the data to the receiver which communicates the values to the ThingSpeak Cloud via Wi-Fi. At this time, the super-capacitor through the PMU delivers the high current needed by the transmitting module. Therefore, the readings of the STEH are saved, and available to visualize and analyze as live data streams. Figure 9 illustrates this communication process by means of flowchart analysis and a pictorial representation of different stages involved. The receiving ESP32 LoRa module with antenna within a smart building is shown in Figure 10.

### 3.4. Laboratory Experiments

The experimental set-up of the HCP-STEH is shown in Figure 11a,b. The experimental procedure involved simulating wind by using an air blower to direct a beam of wind to the HCP ventilator turbine.

Here, a laser non-contact digital tachometer was used to measure the rotating speed of the pinwheels. The tachometer works best on a reflective surface, however, since there are multiple blades of the HCP, and to enhance the measurability and accuracy of the laser tachometer, reference has to be made to one-blade rotation. Hence, the blades were painted black, and a reflecting tape was added to a single blade of the HCP in order to obtain tachometer reading of its speed of rotation as shown in Figure 11b.

The SWDT-EH was tested at low and high wind speeds between 0.6 km/h and 16 km/h. The wind speed and the rotational speed (rpm) of the pinwheels were recorded against the output voltage from the generator which was observed with the aid of a digital oscilloscope and a voltmeter. Results obtained were analysed using MATLAB.

### 3.5. Rooftop Experiments

The Cloud-based remote monitored HCP-STEH was set up on a building with a flat rooftop within the University of Aveiro, Portugal.

The LoRa transmitter sends the voltage and current readings obtained to the LoRa receiver every 15 min, while being on sleep mode during the intervals to conserve energy. The HCP-STEH setup remained on the rooftop and the voltage and current readings were monitored both in the laboratory and via the Cloud for five days consecutively. Figure 12 shows a voltage of approximately 4.1 V available for charging the super capacitor as the output of the HCP-STEH at a local average wind speed of 3.7 m/s, in one of the five days that the harvester remained on the rooftop. This is sufficient to power an IoT power supply pin of 3.3 V.

## 4. Results and Discussions

### 4.1. Laboratory Experiments

The output voltage waveform (at no load and unfiltered) for certain different wind speeds are as shown in Figure 13.

Here, the figure shows that the output voltage is DC with fluctuations, however, a relatively clean voltage waveform was obtained to supply the connected IoT devices from the PMU.

The scattered plot of the no-load peak output voltage from the HCP-STEH with wind speeds is as shown in Figure 14.

The graph shows that the peak output voltage increases with wind speed. At very low wind speeds, the output voltage is low, and at about a wind speed from 3 km/h, the output voltage increases significantly with wind speed. This region shows a linear relationship between output voltage and wind speed which agrees with the expectations of Equation (Equation 14). At about a wind speed of 16 km/h, the generated peak voltage is about 16 V.

Careful inspection of Figure 14 reveals that the peak output voltage increases with an increase in wind speed but in a slightly non-linear fashion at very low wind speeds, and towards the high wind speed values. This is accentuated by the fitted red curve on the data shown in Figure 15.

The fitted curve shows that at low wind speeds, low voltage is produced, and there is a non-linear rise in output voltage with wind speed. This shows that even with a relatively light or gentle breeze, the harvester has the capability of generating output voltage. From a wind speed of about 4 km/h, output voltage increases significantly with an increase in wind speed in a linear fashion; and the generated output voltage is sufficient to operate low-power IoT devices. Additionally in this region, a small percentage increase in wind speed gives a large percentage increase in the generated output voltage. At high wind speeds above 14 km/h to 16 km/h, the peak output voltage rises somewhat non-linearly, and seems to approach a limiting value above 16 V. This gives the operational output voltage range of the harvester.

The wind speed directly affects the rate of change of magnetic flux in the electromagnetic converter, and in turn, the amount of electromotive force (emf) realisable. The speed is therefore an important factor in harvesting the energy, and it can be aided by the height of mounting of the turbine. This makes the rooftop upon which the chimney caps are usually mounted an added advantage.

### 4.2. Rooftop Experiments with Cloud-Based Monitoring

The harvester was mounted on a rooftop, and monitored for 5 days as pointed out in Section 3. Voltage and current data from the harvester was received and saved to the Cloud from 15:45 of Day 1 to 17:20 of Day 5. Figure 16, Figure 17, Figure 18, Figure 19 and Figure 20 show the output voltage and current from the STEH received for Day 1 to Day 5, respectively at intervals of 15 min. The voltage and current data depends not only on the wind speed but also on the load requirement at the time of measurement.

-From Day-1 data presented in Figure 16, due to variability of wind speed, different values of voltage generated and current can be observed. For about an hour after setup and monitoring, relatively moderate levels of measured generated voltage can be observed; above 1 V with maximum of 2.8 V; which surprisingly produced a constant level of current around 3.3 mA for a while. Thereafter, as a result of reduced wind speed, the measured output voltage decreased for the rest of the day; lower than 1 V, and this reduction is also applicable to the current values.-In Figure 17, corresponding to Day 2 of the observation, it was observed that the measured output voltage was around 1 V, except at some instances where a maximum voltage of 9 V and maximum current of 5 mA were measured. Here it can be deduced that it was a calm day with the harvester on the rooftop.-Day 3 of the observation is as indicated by Figure 18 where the received output voltage was around 0.5 V. At a particular instance in the early hours of the day, a voltage of 4 V was obtained.-On Day 4 of the observation, as shown in the graph of Figure 19, the measured output voltage varied within 0.3 V and 0.75 V. The current varies in similar fashion as the voltage.-Day 5 measured output voltage was around 0.5 V, as shown in Figure 20. There were some instances where a significant amount of generated voltage was measured as 3 V, 6 V, and 8 V, with some instances of current ranging between 4 mA and 6.4 mA. Judging from the received measured output voltage from the STEH for 5 days, it can be deduced that the average wind energy received was sufficient to power low-energy-consuming IoT devices, irrespective of whether it was a very windy or calm day with the harvester on the rooftop. It is noteworthy that many of these HCP are scattered abroad on Portuguese rooftops, and some parts of Europe, hence, it is a worthy effort to venture into these ambient energy resources in order to solve the energy needs of modern smart city buildings.

Figure 21 shows the measured output voltage of the HCP-STEH for the entire five days with an 8-point moving average trend-line superimposed. Since the time intervals are 15 min therefore, 8-point average would give average values of 2 h intervals.

Figure 22 shows the wind speed in Aveiro for the month of September 2022 obtained from [54], for comparison purposes. The figure shows the maximum wind speed; the minimum wind speed; and the average wind speed. The five days of measurement and monitoring of the HCP-STEH on the rooftop corresponds to September 20th to 24th indicated within the red rectangle. The time interval of values of wind speed shown in Figure 22 is 6 h, while those from the harvester are at a 15 min interval. However, the local wind speed around the harvester on the rooftop would be affected by the surrounding structures and topography which might make its values slightly or significantly different from the values in Figure 22.

Figure 23 shows a snapshot of “ThingSpeak” analytic Cloud platform of the HCP-STEH Cloud-based output data for part of Day 2 and the whole of Day 3, while Figure 24 shows the snapshot for the latter part of Day 3 and the whole of Day 4.

A close similarity can be obtained by observing Figure 23 and Figure 24 in comparison with Figure 18 and Figure 19. Though the data are saved on the Cloud platform at intervals of 15 min, some missing data observed in Figure 23 are at the instances when the transmitted data failed to be saved on the Cloud platform. This manifests as some portions of irregular data intervals on the time axis. However, there were no missing data in Figure 24 corresponding to the Day 4 observation of “ThingSpeak”.

## 5. Conclusions

The home chimney pinwheels (HCP) for the exhaust outlet on most Portuguese home rooftops have been explored in this research as a wind Smart Turbine Energy Harvester (STEH) to harvest wind energy by taking advantage of the altitude and the free ambient energy that is made available by the set-up from their continuous effortlessly spinning on rooftops. The device is passive in most home chimney towers, whereas they continually spin at the slightest provocation of wind. The electromagnetic converter employed has a high power-to-weight ratio, high efficiency, low maintenance, and is of low cost.

In indoor and outdoor experiments, an output voltage of 0.3 V to 16 V was realised for a wind speed between 0.6 to 16 km/h, which is sufficient to operate low-power IoT devices deployed around a smart city. The harvester’s output energy data was remotely monitored via the IoT analytics Cloud platform “ThingSpeak” by means of LoRa transceivers, which also obtain supply from the harvester via its power management unit.

The overall goal is the adaptation of the existing and commonplace roof ventilator turbines with a simple low-cost electromagnetic converter to bring about altogether low-cost wind STEH with easy installation and minimal maintenance for smart city low power ubiquitous devices. The HCP can be a battery-less “stand-alone” low cost STEH, with no grid connection, and can produce energy that is ubiquitous for industrial process monitoring as well as for other IoT application in homes. The device can be installed as attachments to IoT or wireless sensors nodes (WSN) in smart buildings and cities, thereby enabling them to scavenge their own energy.

Smart cities usually employ the use of IoT, IIoT, or WSN to gather physical information from the immediate environment.

In addition, the smart building rooftop is a viable location for IoT, or any other autonomous device to receive physical data from the surrounding. This informs the type of energy harvesting phenomena that are applicable in the case of smart buildings. Here, the use of solar harvesting, thermal, or wind vibration are the energy forms that are easily accessible from building tops. However, in order to obtain a 24 h, constant and ubiquitous energy supply, the use of STEH is presented in this work, since the solar harvesting output is limited to only daytime, and thermal harvesters are likewise limited for similar reasons. Hence, STEH is presented as having been experimented to produce energy at a very low input of wind energy to moderate speed. In addition, the activities of this method of harvesting wind energy was autonomously monitored and can be remotely controlled to create an Internet of Things network of making an “energy-speak”.

## Figures and Tables

**Figure 1 sensors-23-02858-f001:**
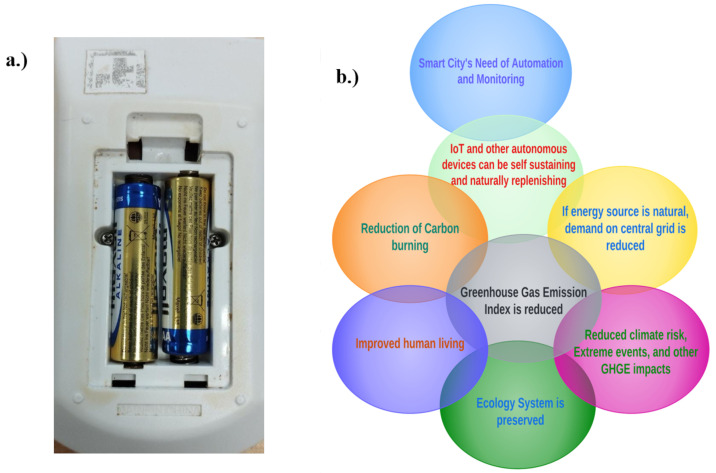
Motivations and ecological impacts of STEH: (**a**) Problems of Battery Replacement. (**b**) Motivation for STEH.

**Figure 2 sensors-23-02858-f002:**
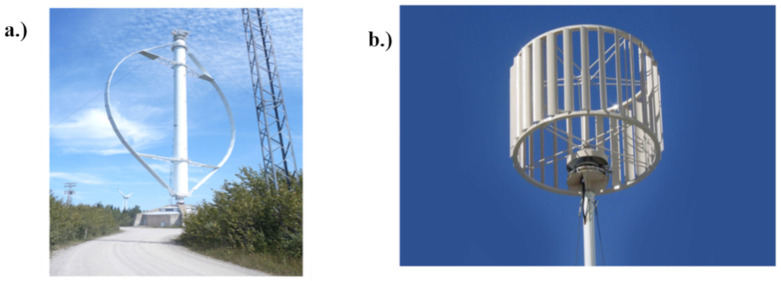
(**a**) Vertical Axis Wind Turbine (VAWT). (**b**) Multiple-blade VAWT [38,40].

**Figure 3 sensors-23-02858-f003:**
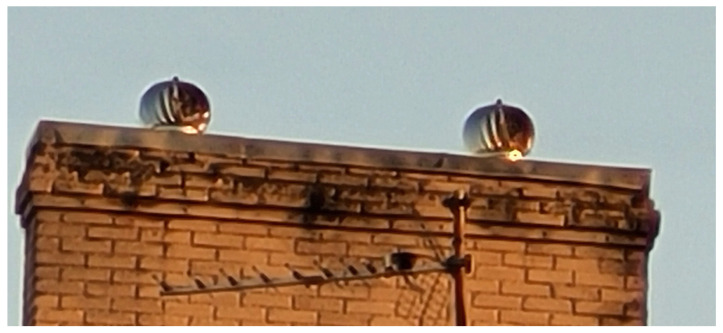
Home chimney pinwheels (HCP).

**Figure 4 sensors-23-02858-f004:**
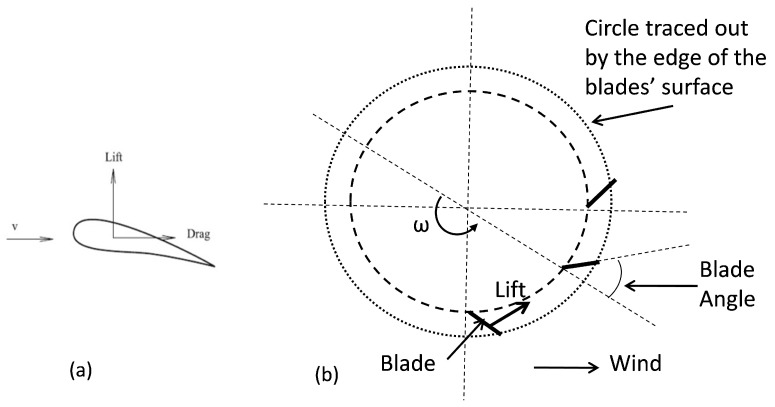
(**a**) Aerodynamic lift and drag acting on a blade. (**b**) Mid-horizontal cross-sectional view of the turbine and wind flow.

**Figure 5 sensors-23-02858-f005:**
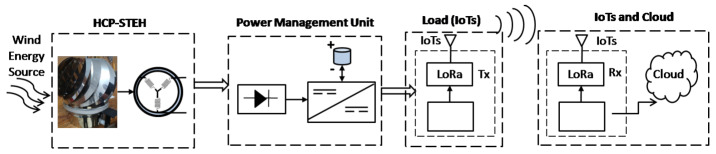
Block Diagram of energy harvesting mechanism and remote monitoring of Wind STEH.

**Figure 6 sensors-23-02858-f006:**
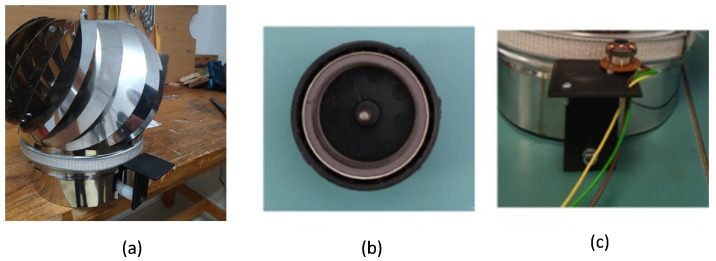
STEH fabrication from Home Chimney Pinwheels (HCP): (**a**) Physical outlook, (**b**) the magnetic cap of the converter, (**c**) the harness showing the converter winding coils with the magnetic cap uncovered.

**Figure 7 sensors-23-02858-f007:**
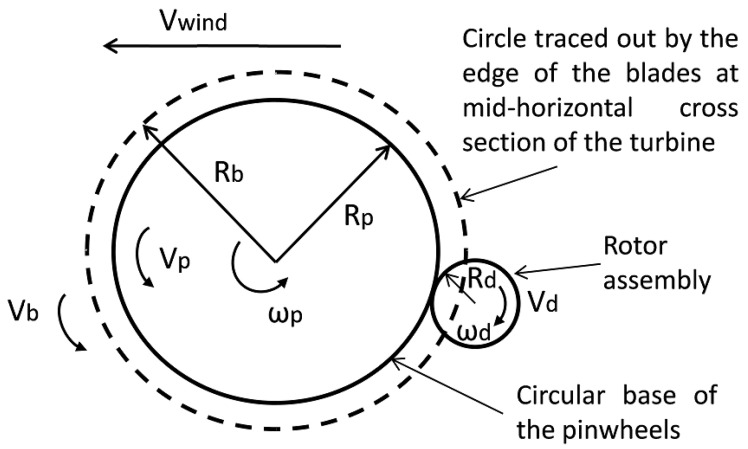
STEH-HCP Top-view rotation.

**Figure 8 sensors-23-02858-f008:**
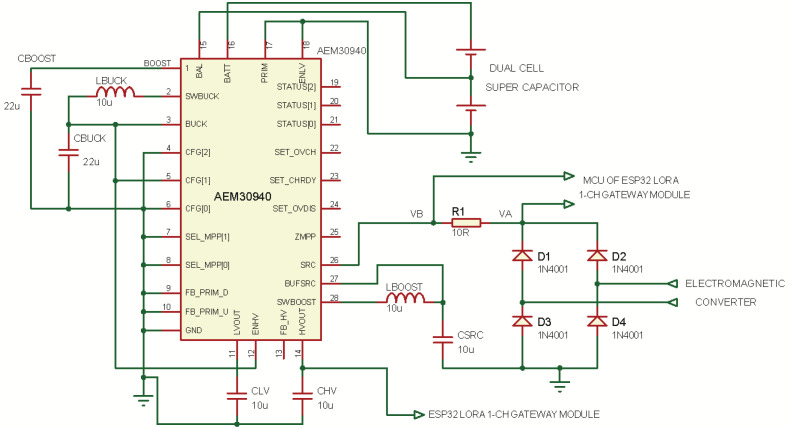
PMU Circuit diagram.

**Figure 9 sensors-23-02858-f009:**
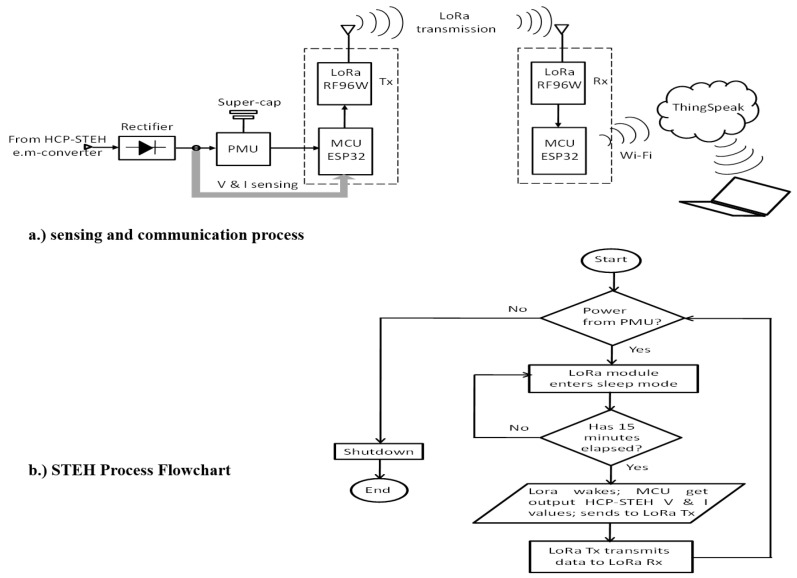
STEH Smart Sensing and Communication Process: (**a**) Sensing and communication (**b**) Flowchart.

**Figure 10 sensors-23-02858-f010:**
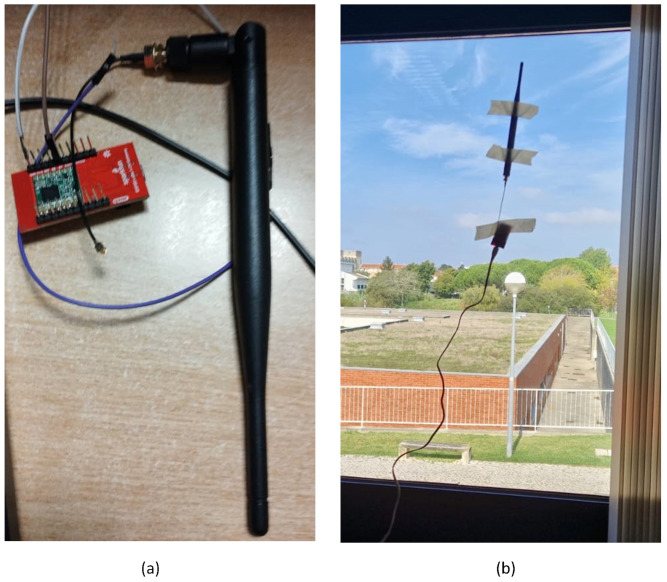
The ESP32 LoRa 1-CH Gateway Receiving Module (**a**) with a duck antenna, (**b**) mounted within a building.

**Figure 11 sensors-23-02858-f011:**
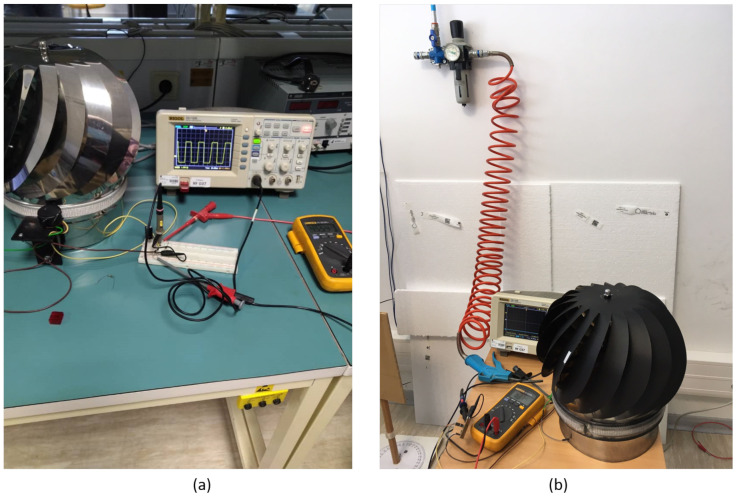
Laboratory Set-Up for the HCP Smart Turbine Energy Harvester. (**a**) Experimental set-up, (**b**) Set-up for wind measurement with output voltage.

**Figure 12 sensors-23-02858-f012:**
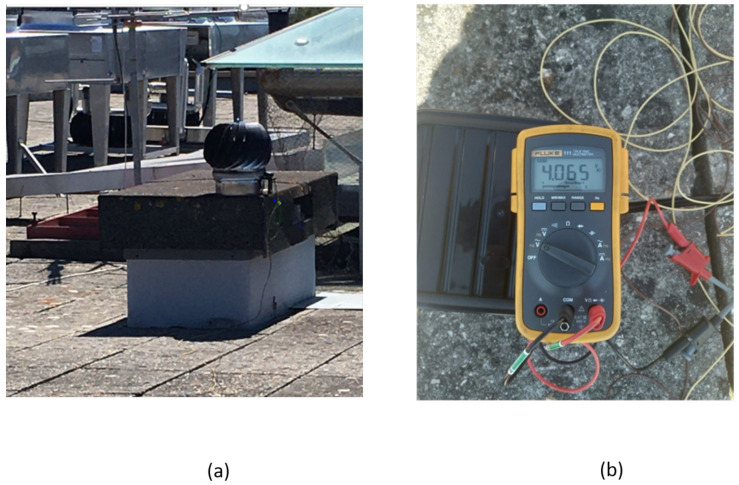
STEH Rooftop Set-up: (**a**) HCP-STEH on flat rooftop (**b**) output reading.

**Figure 13 sensors-23-02858-f013:**
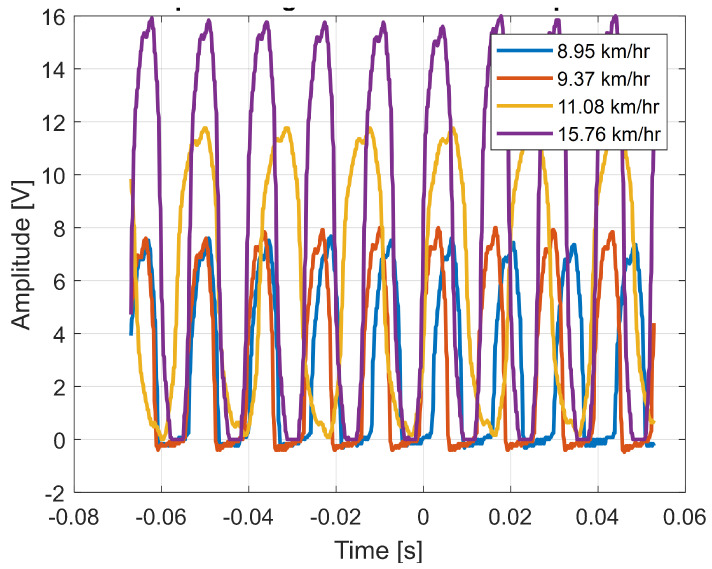
Output voltage waveforms for different wind speeds.

**Figure 14 sensors-23-02858-f014:**
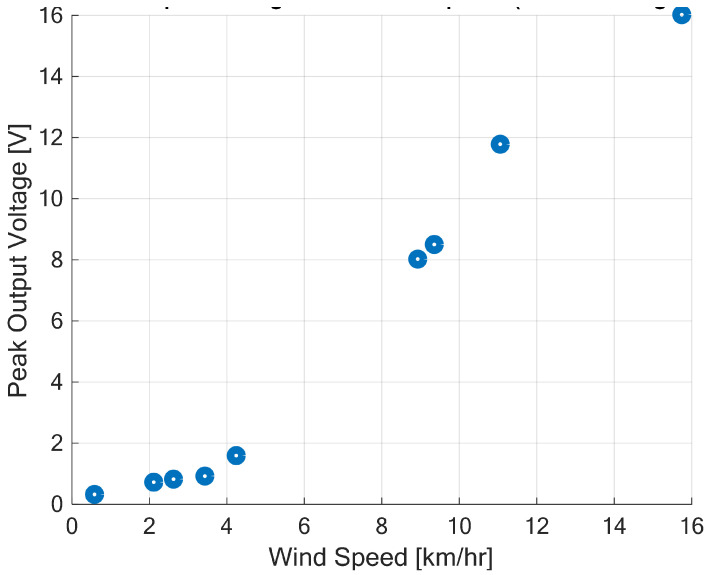
No-load peak output voltage with wind speeds.

**Figure 15 sensors-23-02858-f015:**
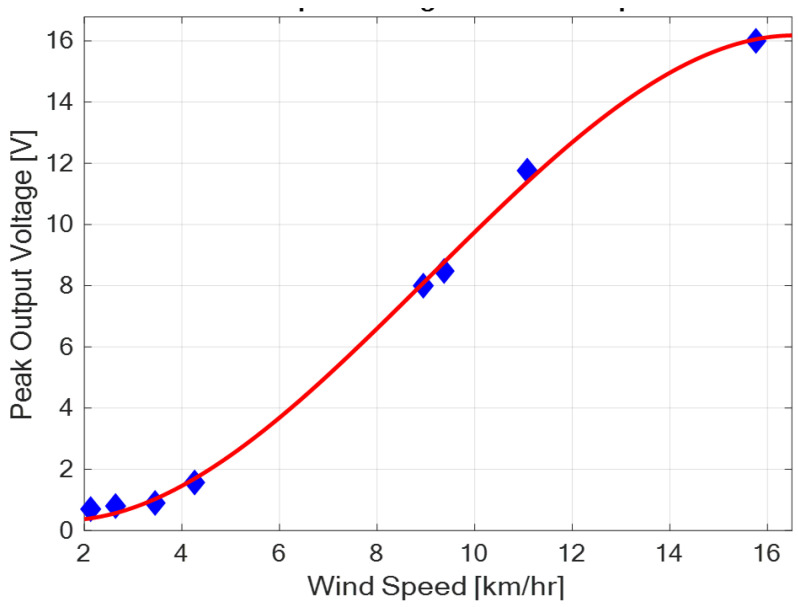
Fitted output voltage curve with wind speed.

**Figure 16 sensors-23-02858-f016:**
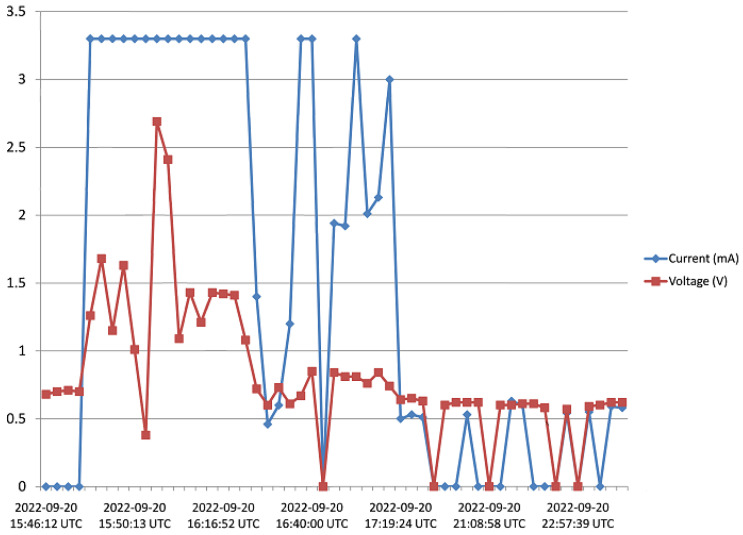
Recorded output data for Day 1.

**Figure 17 sensors-23-02858-f017:**
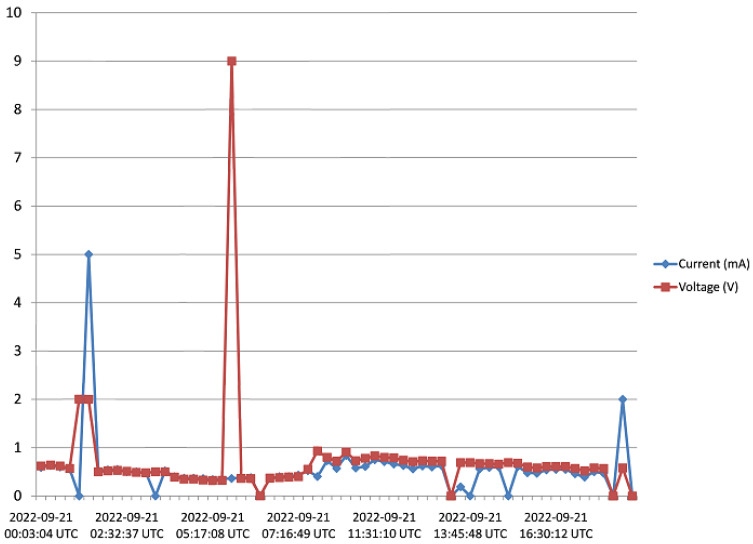
Recorded output data for Day 2.

**Figure 18 sensors-23-02858-f018:**
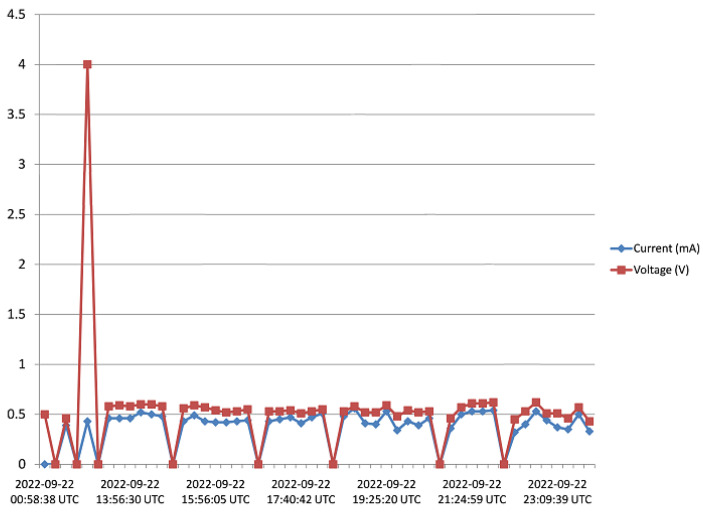
Recorded output data for Day 3.

**Figure 19 sensors-23-02858-f019:**
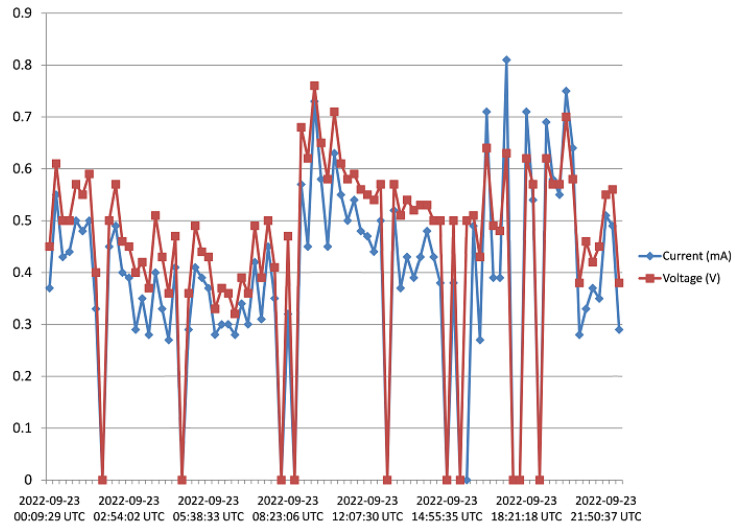
Recorded output data for Day 4.

**Figure 20 sensors-23-02858-f020:**
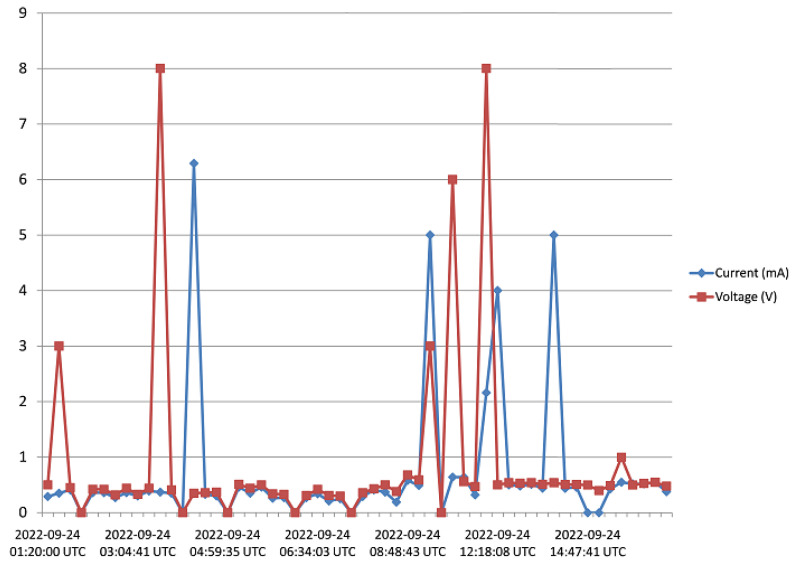
Recorded output data for Day 5.

**Figure 21 sensors-23-02858-f021:**
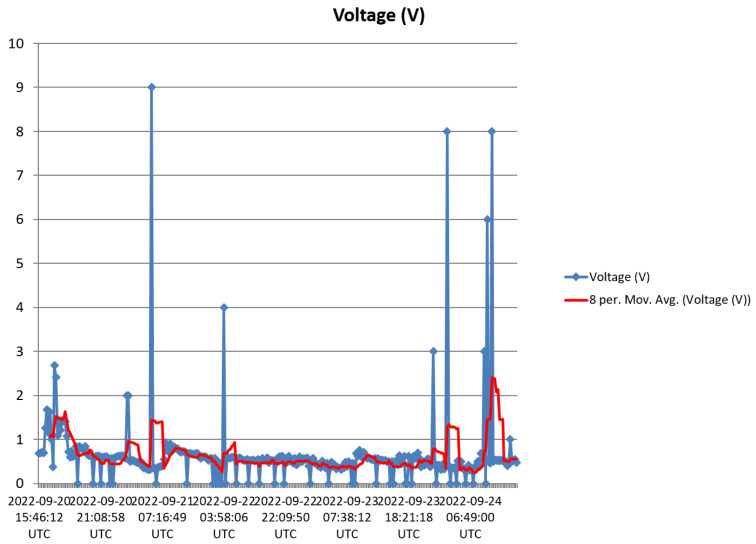
Measured output voltage of the harvester for the five days with an 8-point moving average trend line.

**Figure 22 sensors-23-02858-f022:**
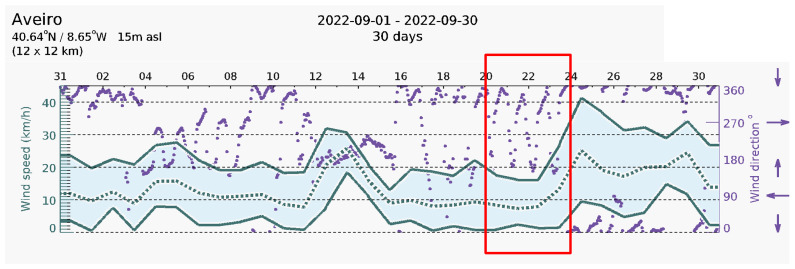
Wind speed in Aveiro for the month of September 2022. [Source: www.meteoblue.com, accessed on 12 December 2022]. The five days of monitoring of the harvester is indicated by the red rectangle.

**Figure 23 sensors-23-02858-f023:**
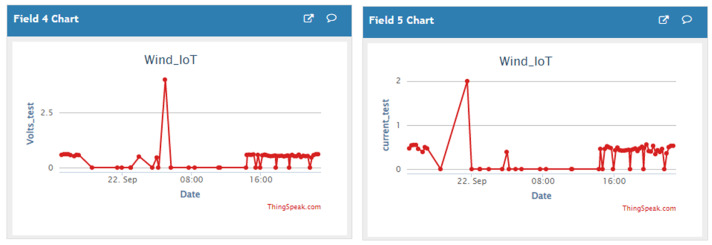
A snapshot of the HCP-STEH Cloud-based output data for Day 3 on the “ThingSpeak” Cloud platform.

**Figure 24 sensors-23-02858-f024:**
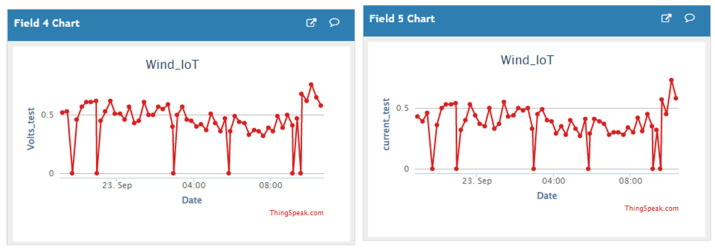
A snapshot of the HCP-STEH Cloud-based output data for Day 4 on the “ThingSpeak” Cloud platform.

## Data Availability

Not Applicable.

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
