# Peer review of "Home Chimney Pinwheels (HCP) as Steh and Remote Monitoring for Smart Building IoT and WSN Applications†"

_sensors, 2023, doi:10.3390/s23052858_

Round 1
Reviewer 1 Report
Fig 1. is just generic image from the internet, with no significant meaning. I suggest authors to draw they own image.
Fig 5. is too simplified. Authors should provide schematics for each block. Also, paragraph 3.2 and 3.3 are giving very little information about the used circuit in this paper.
Fig 14 - 19 : there is no need for so many timestamps on x axis.
Author Response
Please find response here attached.

Reviewer 2 Report
HOME CHIMNEY PINWHEELS (HCP) AS STEH AND REMOTE MONITORING FOR SMART BUILDING IOT AND WSN APPLICATIONS
1. Starting from the introduction section, research contributions should be highlighted in bullet points.
2. Section 3.2 power management unit(PMU) should need revision with proper diagram etc.
3. Section 3.3 Smart sensing and cloud platform need revision with more detailed illustrations with diagrams etc.
4. Section 3.4 laboratory experiments need major revisions with precise more detailed justifications.
5. Section 3.5 Rooftop experiments need major revisions with more technical details.
6. Manuscript requires one section for limitations and future research directions.
7. One detailed flow chart is needed for highlighting the research contributions clearly.
8. Complexity analysis should be added in this manuscript.
9. Figure 18, 20,21 and 22 needs detailed justifications.
Author Response
Please find response here attached.

Round 2
Reviewer 1 Report
no comments
Reviewer 2 Report
HOME CHIMNEY PINWHEELS (HCP) AS STEH AND REMOTE MONITORING FOR SMART BUILDING IOT AND WSN APPLICATIONS
The current state of the manuscript may be considered for the publications since most of the issues are addressed by the authors.